# Effects of Organic Amendments on Soil Pore Structure under Waterlogging Stress

**Kefan Xuan** [1,2] [iD], **Xiaopeng Li** [1] [iD], **Jiabao Zhang** [1], **Yifei Jiang** [1,2], **Bin Ma** [3,4,5] and **Jianli Liu** [1,*]

1   Department of Soil Physics and Saline Soils, Institute of Soil Science, Chinese Academy of Sciences, Nanjing 210008, China
2   College of Advanced Agricultural Sciences, University of the Chinese Academy of Sciences, Beijing 100049, China
3   Institute of Soil and Water Resources and Environmental Science, College of Environmental and Resource Sciences, Zhejiang University, Hangzhou 310058, China
4   Zhejiang Provincial Key Laboratory of Agricultural, Resources and Environment, Zhejiang University, Hangzhou 310058, China
5   ZJU-Hangzhou Global Scientific and Technological Innovation Center, Zhejiang University, Hangzhou 310058, China
*   Correspondence: jlliu@issas.ac.cn; Tel.: +86-025-8688-1226

**Abstract:** Organic amendments are a proven method of reducing soil erosion. However, the effect of organic additives on the pore properties of soils waterlogged by extreme rainfall has been minimally investigated. In this study, we collected rainfall data, established a field experiment consisting of randomized groups, and imaged the pore structure of waterlogged soil treated with different organic amendments (9 t ha$^{-1}$ of maize straw [MS], 2.37 t ha$^{-1}$ of cattle manure [OF], a mixture of 9 t ha$^{-1}$ of MS and 1.89 t ha$^{-1}$ of cattle manure [SOF], 7.4 t ha$^{-1}$ of biochar [BC], 8.1 t ha$^{-1}$ of woody peat [WP], and 3 L ha$^{-1}$ of polyacrylamide [PAM]) in three-dimensions by X-ray microtomography and digital image analysis, which further quantified the effects. The results showed that, compared with the control, BC increased the total porosity by 54.28% and the connected porosity by 119.75%, but did not affect the pore shape and size distribution. BC and SOF improved the soil connectivity indexes; e.g., their C/I ratios increased by 177.44% and 149.62%, and the coordination numbers increased by 6.75% and 15.76%, respectively. MS had fewer, but longer and larger, channels and throats. Extreme precipitation events were significantly negatively correlated with all connectivity indicators. This study shows that organic materials can optimize the pore structure of waterlogged soil, with BC being the most resistant to erosion. However, extreme precipitation events can counteract the benefits organic additives have on soil pore structure.

**Keywords:** organic amendments; soil pore structure; extreme precipitation events; biochar

## 1. Introduction

Extreme precipitation events occur frequently and have been among the most serious weather-related hazards since meteorological records began [1]. Large amounts of precipitation falling over a short period can influence soil function, affecting soil water distribution, inhibiting root respiration and seed germination [2–4]. Nevertheless, soil function depends on soil structure, and while the above-mentioned adverse consequences due to waterlogging are closely related to soil structure, few studies have investigated the effects of extreme precipitation on soil physical properties, especially pore structure.

Most of the current related studies are conducted within the laboratory, that is, by simulating the effects of raindrops on the topsoil. Rainfall droplets strike the surface soil, resulting in the fragmenting of soil aggregates, which block soil pores and affect cation exchange. These processes lead to the formation of a denser and relatively thin layer on the soil surface (i.e., physical crusting) [5–7]. The dispersed soil particles block the

pores and change the surface soil's pore structure, leading to poor hydraulic conductivity and permeability, a reduced infiltration rate, increased surface runoff, and intensified erosion [8,9]. Yang et al. [10] demonstrated that the clogging ratios caused by raindrops increased with rain intensity and rainfall energy. Rousseva et al. [11] concluded that rain can affect soil macroporosity to a depth of at least 6 cm and that the reduction in porosity may be due to the pressure wave generated by the impacts of raindrops at the soil surface. Research on the mechanisms by which raindrops affect soil surface structure has come a long way, but little has been reported on how to avoid or mitigate rainfall damage to the pore structure.

Many studies have demonstrated that organic materials can reinforce soil structure, reduce soil erosion and facilitate crop growth. The introduction of woody peat and rotten straw could reduce soil particle size and bulk density and alleviate issues of soil viscosity [12]. The combined application of inorganic fertilizers, green manure, and swine manure optimized soil pore structure, improved soil aggregation, reduced nitrogen (N) loss, and increased wheat grain yield [13,14]. Peng et al. [15] reported that swine manure and straw mulch could reduce soil erosion. Polyacrylamide was effective in promoting soil aggregation, controlling soil runoff and reducing soil loss on erosion slopes [16]. In addition, based on an indoor experimental study, Fei et al. demonstrated that biochar and polyacrylamide can improve the water retention capacity as well as structural stability of saline soils [17]. Biochar is an important embodiment of solid waste resource utilization, with various production processes (e.g., pyrolysis, gasification, and hydrothermal carbonization), resulting in many products with different properties, most of which can be used as soil conditioners to alleviate the stresses present in the soil, such as salinity, water stress and hydrocarbon contamination [18–20]. Zheng et al. showed that severe water stress can affect peanut growth, but biochar-based fertilizer can mitigate this negative effect [21]. Biochar application alleviated rice salt stress by improving soil nutrient conditions, increasing bacterial abundance and changing microbial community structure [22]. Regarding biochar remediation of hydrocarbon-contaminated soils, Dike et al. [23] showed that biochar promotes the degradation of petroleum hydrocarbons in soil mainly by affecting soil microorganisms and is more effective when applied in combination with nutrient fertilizers. Biochar increased the total porosity and pore volumes by over > 5 µm within soils, and pore spaces >5 µm had positive effects on microbial diversity and abundance [24]. The artificial neural network developed by Garg et al. [25] showed that an amendment containing 5% biochar was the best for reducing soil erosion. Nevertheless, few studies have focused on quantifying microscopic pore structure or whether organic materials can still optimize soil structure after erosion caused by natural rainstorms.

Given the above information, we combined the results of previous studies and collected in situ soil samples from a test site that had experienced extreme rainfall. We used X-ray microtomography as well as advanced digital image processing methods to visualize and quantify the effects each organic material had on the soil pore structure under waterlogging stress. Based on this, the relationship among extreme rainfall events, organic amendments (i.e., maize straw, biochar, and manure etc.), and soil pore characteristics were further explored in an attempt to select the best materials for erosion resistance.

## 2. Materials and Methods

### 2.1. Site Description and Experimental Design

The study was conducted in October 2018 at the Fengqiu State Key Agro-Ecological Experimental Station, Chinese Academy of Sciences (35°00′ N, 114°24′ E) in Fengqiu, Henan Province, China. The soil is classified as Calcaric Fluvisol according to the FAO (Food and Agriculture Organization of the United Nations, Rome, Italy) and has a sandy loam texture consisting of 52% sand, 33% silt, and 15% clay [26]. The area is semiarid and has a warm temperate continental monsoon climate with a mean annual temperature of 13.5 ~ 14.5 °C, and the mean annual precipitation is 615 mm (mainly occurring from July to September). Other initial soil properties of the site are given in Table 1 as follows.

**Table 1.** Initial soil characterization of the site.

| Initial Soil Characterization | Average Value |
|---|---|
| Bulk density (g cm$^{-3}$) | 1.42 |
| pH | 8.63 |
| Organic C (g kg$^{-1}$) | 6.09 |
| Total N (g kg$^{-1}$) | 0.46 |
| Total P (g kg$^{-1}$) | 0.52 |
| Total K (g kg$^{-1}$) | 18.08 |
| Cation exchange capacity (cmol kg$^{-1}$) | 8.01 |

The cropping system in the trial area was a winter wheat and summer maize rotation. Field trials were conducted in a randomized group design with seven treatments in triplicate: (a) control, with no organic amendments incorporated (CK); (b) maize straw, totaling approximately 9000 kg ha$^{-1}$, crushed from the straw of the previous maize season, about 2–3 cm in size (MS); (c) organic fertilizer, which consisted of fermented cattle manure with a 14% organic matter and 0.9% total N content (OF); (d) biochar, provided by Qinfeng Zhongcheng Biomass New Materials (Nanjing, China) Co., Ltd., is a high-temperature carbonization product of maize straw, with pyrolysis temperature of 450–500 °C and carbonization time of 120 min, which has high specific surface area and porosity, containing 49% organic carbon (BC); (e) woody peat, supplied by Yantai Ganzhiyuan Biotechnology Co., Ltd., accumulated and transformed from plant residues in an environment with reduced waterlogging, containing 48% organic carbon and 76% humic acid (WP); (f) an anionic polyacrylamide with high molecular weight (PAM); and (g) maize straw mixed with organic fertilizer (SOF). The plot size was 25.92 m$^2$. In this experiment, the N content was set at 230 kg·ha$^{-1}$ and the C content of the MS treatment was used as a benchmark to determine the specific amount of organic material to be applied to each plot to maintain consistency. Specific fertilizer application rates are given in Table 2, with further details already reported by Xuan et al. [27]. Other field management practices were kept consistent among treatments.

**Table 2.** Annual amounts used in the experimental treatments.

| Treatment | N (kg ha$^{-1}$) | P (kg ha$^{-1}$) | K (kg ha$^{-1}$) | Maize Straw (kg ha$^{-1}$) | Organic Fertilizer (kg ha$^{-1}$) | Biochar (kg ha$^{-1}$) | Woody Peat (kg ha$^{-1}$) | PAM (L ha$^{-1}$) |
|---|---|---|---|---|---|---|---|---|
| CK | 230/230 [1] | 130/130 | 120/120 | -/- | -/- | -/- | -/- | -/- |
| MS | 230/230 | 130/130 | 120/120 | 9000/- | -/- | -/- | -/- | -/- |
| OF | -/230 | 130/130 | 120/120 | -/- | 23,662/- | -/- | -/- | -/- |
| BC | 230/230 | 130/130 | 120/120 | -/- | -/- | 7400/- | -/- | -/- |
| WP | 230/230 | 130/130 | 120/120 | -/- | -/- | -/- | 8100/- | -/- |
| PAM | 230/230 | 130/130 | 120/120 | -/- | -/- | -/- | -/- | 3/- |
| SOF | 211/230 | 130/130 | 120/120 | 9000/- | 1893/- | -/- | -/- | -/- |

[1] The first value is the dosage used during the wheat season, and the second value is the dosage used during the maize season.

### 2.2. Soil Moisture and Meteorological Measurements

To determine the continuous change of soil moisture, a frequency domain reflection (FDR) auto soil moisture monitor with a determination frequency of once an hour was installed in the test site. Precipitation was measured at 8:00 and 20:00 each day using an automatic weather station (M520, Vaisala, Vantaa, Finland), which was also located at Fengqiu State Key Agro-Ecological Experimental Station. This study used the device to collect rainfall for the summer maize growth period (from June to September) since 2018. According to the rainfall grade standard issued by the China Meteorological Administration, precipitation can be classified (by considering both the amount and the duration) as light rain (LR), moderate rain (MR), heavy rain (HR), rainstorm (RS), and large rainstorm (LRS); those with a rainfall rating of HR and above are identified as erosive precipitation events [28].

### 2.3. Soil Sampling and Analysis

After the maize harvest in September 2021, in situ soil samples were collected from each plot using transparent polycarbonate (PC) tubes (diameter 4.6 cm, height 2.6 cm) for analysis from the tilled surface layer (0–20 cm). The 21 soil samples (7 treatments × 3 replicates × 1 sample) collected by the transparent PC tubes were sealed with plastic wrap and stored in a refrigerator at 4 °C for CT scanning.

### 2.4. X-ray CT Scanning and Digital Image Preprocessing

The soil samples collected within the transparent PC tubes were scanned by an industrial nano-CT scanner (Phoenix Nanotom S, Wunsdorf, Lower Saxony, Germany) at the Institute of Soil Science, Chinese Academy of Sciences. The FOD (Focus Object Distance; distance between the soil sample and X-ray source) and FDD (Focus Detector Distance; distance between the X-ray tube and detector) were 200 and 400.03 mm, respectively, corresponding to the 25 μm voxel resolution. The scanning voltage and current were 100 kV and 100 μA, respectively. The sample stage was rotated from 0° to 360° at the same step interval (0.3°), and a total of 1200 projections with a size of 2284 by 2304 pixels were obtained for each soil core that could be used for CT reconstruction. After scanning and reconstruction, approximately 2050 8-bit grayscale images were reconstructed for each sample. Due to the noise baseline and the artifacts on the boundaries of the CT scanning samples, we used ImageJ (V.1.51 W, National Institute of Health, Bethesda, MD, USA) to crop 21 cubes with dimensions of 1200 × 1200 × 1200 voxels (physical dimensions of 3 cm × 3 cm × 3 cm) at the center of each core for the study

### 2.5. Visualization and Quantification of Pore Structures in Soil Cores

The CT images were preprocessed by AVIZO 2021.1 (Thermo Fisher Scientific, Walitham, MA, USA: firstly, the *non-local means filter* module was used to denoise the images, then the *unsharp masking* module was used for edge enhancement, and finally, the *interactive thresholding*, *interactive top hat*, and *or image* modules were applied to threshold segmentation as well as integration of the images to obtain a binary image of the in-situ soil structure.

The software was also used to visualize and quantify the pore structure: the *volume fraction* module was applied to extract the pore volume, the *volume rendering* module was for visualizing the pore structure, the *separate objects* and *label analysis* modules were used to calculate the pore structure related parameters, and the *axis connectivity*, *separate objects*, and *generate pore network* modules were used in combination to generate a pore network model and statistics of the related properties. The specific description of the parameters is shown in Table 3, and more details can be found in Xuan et al. [27].

### 2.6. Statistical Analysis

All statistical analyses were carried out with the SPSS 25.0 software package (SPSS, Chicago, IL, USA). The Kolmogorov–Smirnov test was used to test the normality of data, and the Levene's test was applied to tested the homogeneity of variances. Significant differences in soil structure among treatments were determined by one-way analysis of variance (ANOVA). The differences were determined by least significant difference (LSD) multiple-range tests at the 5% and 1% significance levels. We used heatmap analysis to investigate the correlation of extreme precipitation events and organic management with soil pore structure characteristics, and further grouped the properties using hierarchical clustering based on Euclidean distance. Most of the statistical images were generated by Origin software (OriginLab, Northampton, MA, USA), except for the heat map, which was generated with the *corrplot* package in R software version 4.0.4 (R, Vienna, Austria).

**Table 3.** Parameter descriptions [27].

| Parameter | Formula | | Description |
|---|---|---|---|
| Area3d | $N \times A_0$ | (1) | $N$ is the number of pixels that the pore contains, and $A_0$ is the area of a pixel unit ($\mu m^2$). |
| Volume3d | $N \times V_0$ | (2) | $N$ is the number of pixels that the pore contains, and $V_0$ is the volume of a voxel unit ($\mu m^3$). |
| Eqdiameter | $\sqrt[3]{\frac{6 \times V_{3D}}{\pi}}$ | (3) | |
| Eqradius | $\frac{\sqrt[3]{\frac{6 \times V_{3D}}{\pi}}}{2}$ | (4) | $V_{3D}$ is the volume of each pore ($\mu m^3$). |
| Shape factor | $\frac{\pi^{1/3}(6V_{3D})^{2/3}}{A_{3D}}$ | (5) | $V_{3D}$ is the volume of a pore ($\mu m^3$), and $A_{3D}$ is its surface area ($\mu m^2$). Pores can be divided into three types: regular pores (F $\geq$ 0.5), irregular pores (0.2 < F < 0.5), and elongated pores (F $\leq$ 0.2). |
| Porosity | $\frac{V_P}{V_T}$ | (6) | |
| Connected porosity | $\frac{V_C}{V_T}$ | (7) | $V_P$, $V_T$, and $V_C$ represent the volumes of all pores, the connected pores, and the total volume of the slice, respectively ($\mu m^3$). |
| Isolated porosity | $\frac{V_P - V_C}{V_T}$ | (8) | |
| C/I ratio | $\frac{P_C}{P_I}$ | (9) | $P_C$ and $P_I$ are the porosity of connected and isolated pores. |
| Channel length | $\sum_{i=1}^{n} l_i$ | (10) | $l_i$ is the distance from the pore to the pore center. |
| Tortuosity | $\frac{\sum d(i)}{z(n)}$ | (11) | $d(i)$ is the actual length of the pore between two points along the z-axis and $z(n)$ is the number of slices along the z-axis. |
| Coordination number | | | The number of throats that connect neighboring pores. |

## 3. Results

### 3.1. Characterization of Rainfall during the Maize Season

As shown in Figure 1, in the test area, a total of 133 rainfall events, mainly LR and MR, occurred during the maize seasons from 2018 to 2022. There were 29 erosive precipitation events, 14 in 2021 alone, accounting for approximately half of the total, including 7 HR, 5 RS, and 2 LRS events. The rainfall in 2021 reached a total of 974.9 mm over just four months during the maize season, far exceeding the same period during previous years and exceeding the average annual precipitation of 615 mm for this area.

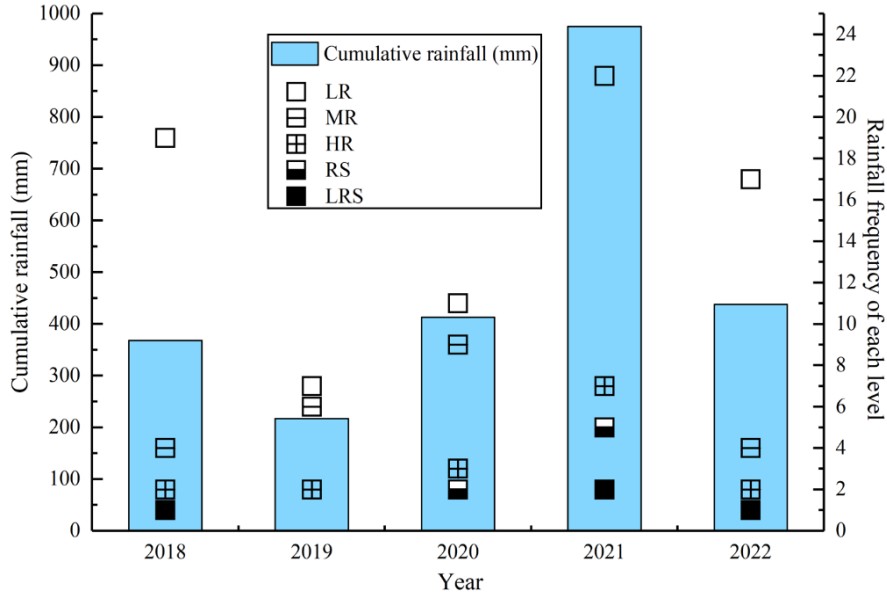

**Figure 1.** Distribution of cumulative rainfall and frequency.

The response process of the average soil moisture in the tillage layer to precipitation during the 2021 maize season is shown in Figure 2. The rainfall contribution from the July to August period (646.5 mm) to the June to September growing season was relatively high in 2021, accounting for 66.31%. The changing patterns of soil moisture and precipitation are the same, but with a slight lag, that is, there is an obvious increase in soil moisture 1 to 2 days after extreme precipitation events. After several LRSs at the heading-flowering stage and filling stage, the average soil moisture was as high as 30%, and in some instances almost reached 90% of the field capacity or even total saturation. The above results show that the growth period of summer maize in 2021 was indeed subjected to waterlogging stresses.

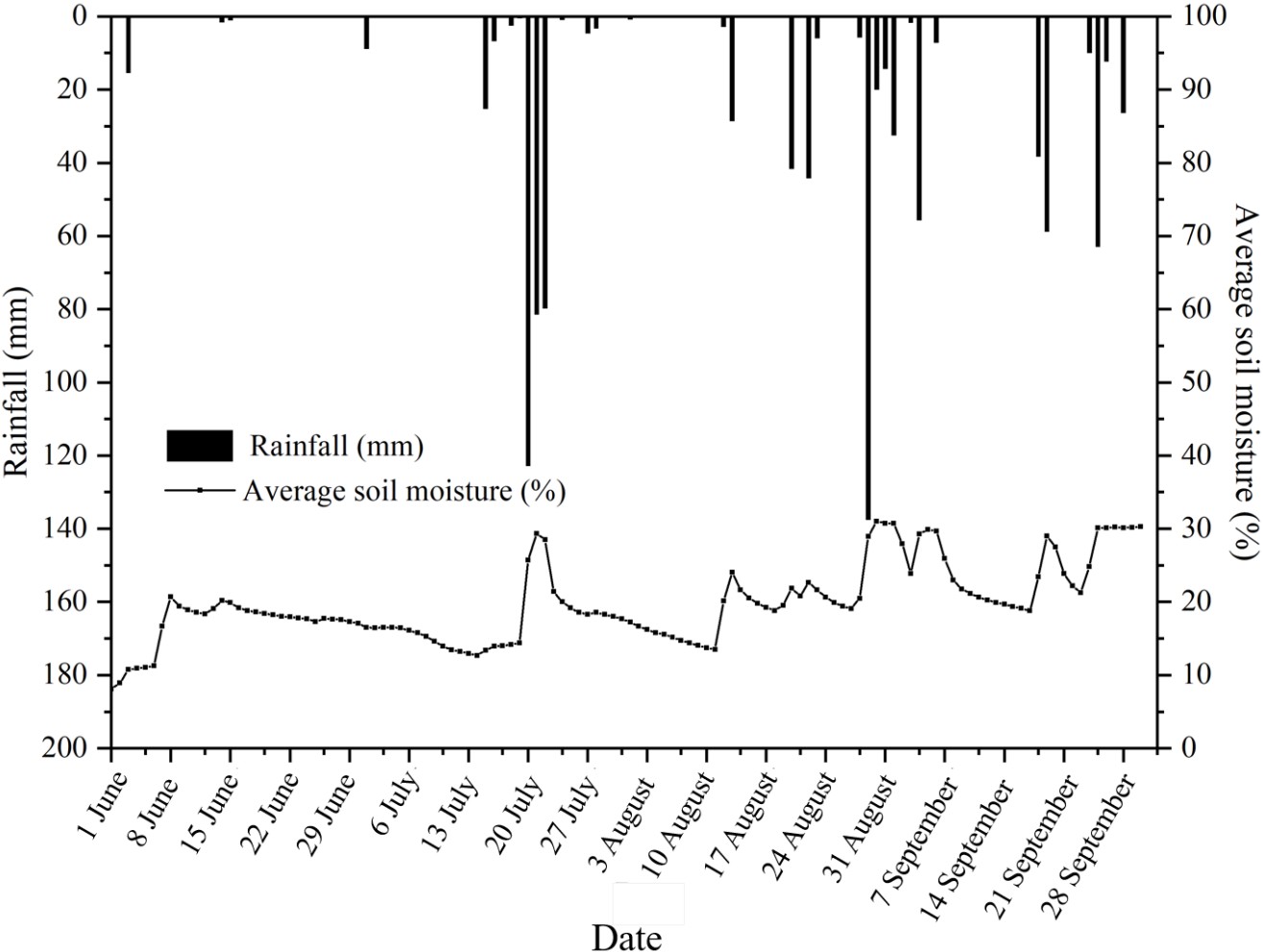

**Figure 2.** Daily rainfall events and soil moisture within the tillage layer in 2021.

*3.2. Visualization and Quantification of the Soil Pore Structure*

Table 4 gives quantitative information on soil pore characteristics for 2020 and 2021. We found that regardless of whether extreme precipitation events occurred, each organic amendment could increase the total and connected porosity and decrease the isolated porosity of the soil to varying degrees, thus increasing the C/I ratio. However, the results in 2021 showed that there was no significant difference among the treatments. The combination of the results of the visual inspection (Figure 3) and analysis parameters showed that BC also performed well after the rainstorm, unlike in 2020 when SOF was the optimal organic material. Compared with CK, BC increased the total porosity by 54.60%, connected porosity by 120.01%, C/I ratio by 177.85%, and reduced isolated porosity by 24.31%.

**Table 4.** Digital image-based calculation of soil porosity and pore shape parameters for 2020 and 2021.

| Pore Properties | Year | Treatments | | | | | | | ANOVA | | |
|---|---|---|---|---|---|---|---|---|---|---|---|
| | | CK | MS | OF | BC | WP | PAM | SOF | Treatments (T) | Year (Y) | T × Y |
| Total Porosity (%) | 2020 | 11.82 ab | 16.40 ab | 11.70 ab | 15.31 ab | 10.45 b | 15.12 ab | 18.27 a | *ns* | *p* < 0.01 | *ns* |
| | 2021 | 4.44 * | 6.49 * | 5.19 * | 6.85 * | 6.23 * | 5.25 * | 6.75 * | | | |
| Connected Porosity (%) | 2020 | 10.16 ab | 15.12 ab | 10.97 ab | 14.77 ab | 9.26 b | 14.49 ab | 17.68 a | *ns* | *p* < 0.01 | *ns* |
| | 2021 | 2.43 * | 4.83 * | 3.51 * | 5.34 * | 4.67 * | 3.60 * | 5.15 * | | | |
| Isolated Porosity (%) | 2020 | 1.66 a | 1.28 ab | 0.73 bc | 0.54 c | 1.18 abc | 0.63 bc | 0.60 bc | *ns* | *p* < 0.01 | *ns* |
| | 2021 | 2.01 * | 1.66 * | 1.68 * | 1.52* | 1.56 | 1.65 * | 1.60 * | | | |
| C/I Ratio | 2020 | 6.30 c | 15.63 abc | 15.29 abc | 29.64 a | 9.02 bc | 25.85 ab | 29.67 a | *p* < 0.05 | *p* < 0.01 | *p* < 0.05 |
| | 2021 | 1.33 * | 3.25 * | 3.05 * | 3.69 * | 3.38 * | 2.04 * | 3.32 * | | | |
| Mean shape factor | 2020 | 1.28 a | 1.26 b | 1.20 c | 1.19 c | 1.19 c | 1.21 bc | 1.20 c | *p* < 0.05 | *p* < 0.01 | *ns* |
| | 2021 | 1.24 | 1.29 | 1.24 | 1.23 | 1.22 | 1.25 | 1.28 | | | |
| Fraction of regular pores (%) | 2020 | 99.25 a | 98.14 ab | 95.56 c | 96.47 c | 97.05 bc | 96.49 c | 95.97 c | *ns* | *ns* | *ns* |
| | 2021 | 96.95 | 98.70 | 95.74 | 95.66 | 96.03 | 98.12 | 98.56 | | | |
| Fraction of irregular pores (%) | 2020 | 0.74 c | 1.85 bc | 4.43 a | 3.522 a | 2.94 ab | 3.50 a | 4.02 a | *ns* | *ns* | *ns* |
| | 2021 | 3.04 | 1.29 | 4.25 | 4.33 | 3.96 | 1.81 | 1.43 | | | |
| Fraction of elongated pores (%) | 2020 | 0.01 | 0.01 | 0.01 | 0.01 | 0.01 | 0.01 | 0.01 | *ns* | *ns* | *ns* |
| | 2021 | 0.01 | 0.01 | 0.01 | 0.01 | 0.01 | 0.07 | 0.01 | | | |

Data for 2020 are cited from Xuan et al. [27]. Different lowercase letters within a row indicate significant differences among treatments, and * indicates significant differences between two years with different rainfall conditions under the same treatment at *p* < 0.05. *ns* represents no statistical significance at the *p* = 0.05 level.

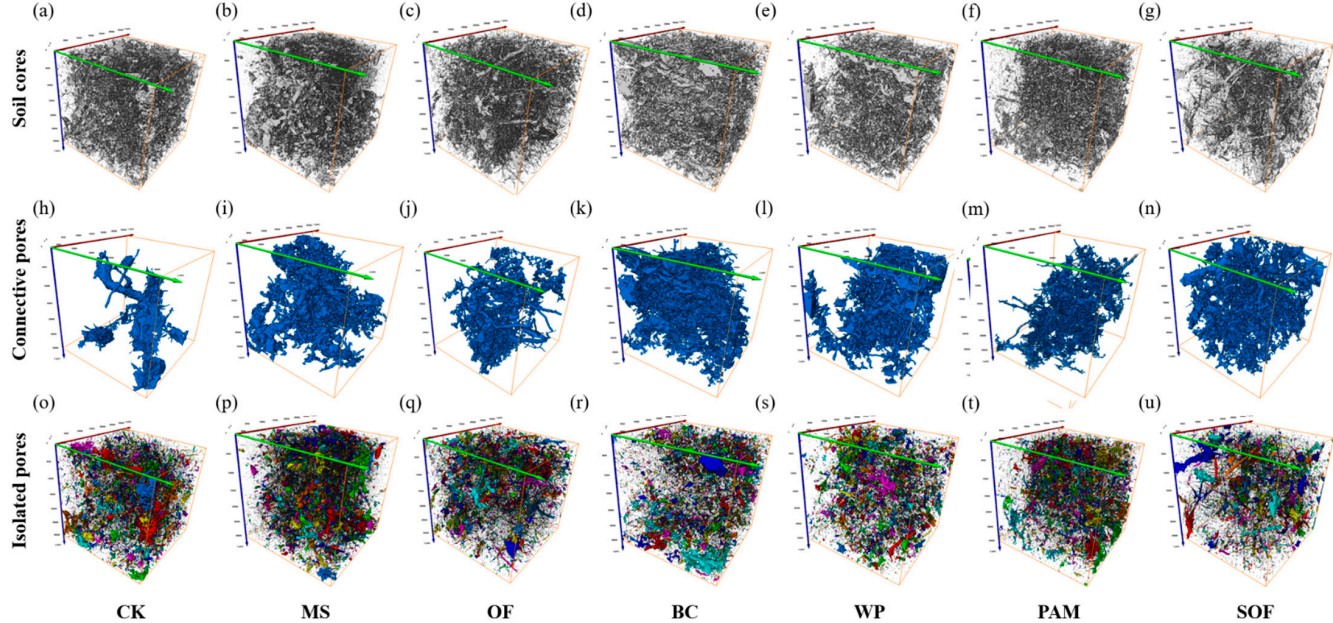

**Figure 3.** Representative 3D visualizations of (**a–g**) soil cores, (**h–n**) connective pores, and (**o–u**) isolated pores from different treatments (physical size = 3 cm × 3 cm ×3 cm).

Compared to 2020, after extreme rainfall, all the parameters showed highly significant changes except for the percentage of each pore shape. Among them, the total porosity, connected porosity, and C/I ratio decreased by 58.40%, 68.06%, and 84.74% on average, respectively, while the isolated porosity increased by 76.52%. It is clear from the ANOVA results that organic management, extreme precipitation, and their interactions can affect the C/I ratio. In terms of pore shape parameters, rainfall can reduce the fraction of regular pores and increase the fraction of irregular pores, but no significant difference was observed. Overall, soil porosity was significantly reduced when placed under waterlogging stress, which was detrimental to soil connectivity, and also altered the pore shape. The application

of organic amendments improved soil pore properties, but this was not evident after the rainstorm.

### 3.3. Soil Pore Distribution

The mean relative proportion of porosity in different treatments was quantified from the CT images (Figure 4). Considering the limitations of image resolution and noise, only soil pores larger than 50 μm were analyzed in this study. The pores were classified into three size classes based on previous studies: micropores (50–500 μm), mesopores (500–1000 μm), and macropores (larger than 1000 μm) [29,30]. Figure 4 shows that almost all pores in each treatment were micropores, with a few mesopores and barely any macropores. The PAM treatment had the most micropores with about 97.74%, while OF had the least micropores with approximately 91.89%. Compared to the control treatment, only OF and BC slightly increased the percentage of mesopores and macropores in the soil. The differences among treatments within each size class were not significant. In general, organic amendments have little effect on soil pore distribution after extreme precipitation events.

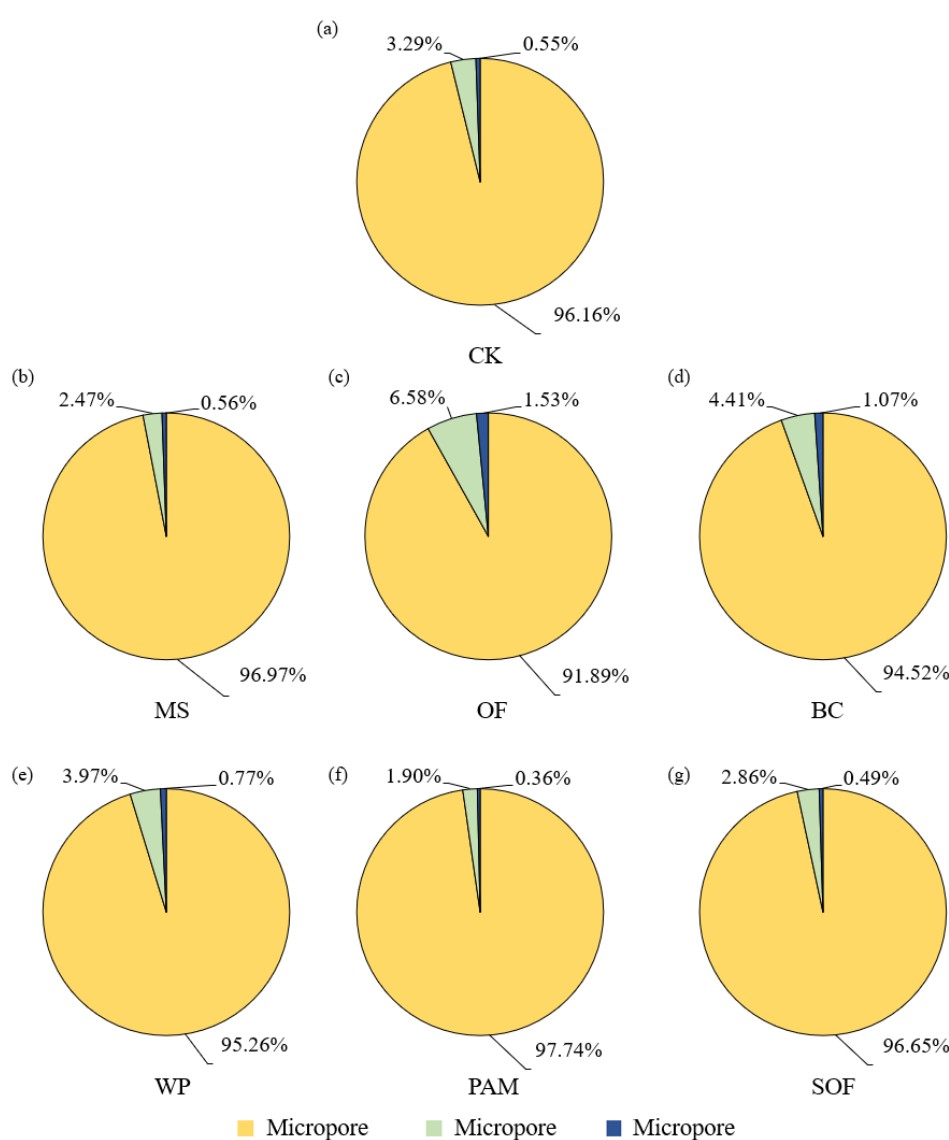

**Figure 4.** The mean relative proportion of porosity in different treatments. (**a**) CK, with control, no organic material incorporated; (**b**) MS, maize straw; (**c**) OF, organic fertilizer; (**d**) BC, biochar; (**e**) WP, woody peat; (**f**) PAM, polyacrylamide; (**g**) SOF, maize straw and organic fertilizer.

### 3.4. Quantification of the Pore Network Model (PNM)

The PNMs (pore network models) shown in Figure 5 demonstrate the connection between pores using a ball-and-stick model. We observed that organic treatments complicated the PNM, especially BC and SOF. For the CK treatment, fewer node pores and lower coordination numbers were indicated by CT scanning, which resulted in a less complex PNM (similar to PAM). The complexity of the pore network observed for the MS and WP treatments was moderate. The characteristics of the PNMs are given in Table 5. SOF had the most node pores and channels (5989 and 10,729, respectively), followed by WP (5523 and 8547), and BC (4164 and 7043). The coordination number is the number of nodes connecting each pore. Larger values indicate that pores are connected to more pores. SOF also had the highest coordination number, which means it had more advantages in terms of connectivity. However, PAM had the lowest coordination number, which was even less than that of CK. For the tortuosity, there were few differences among treatments.

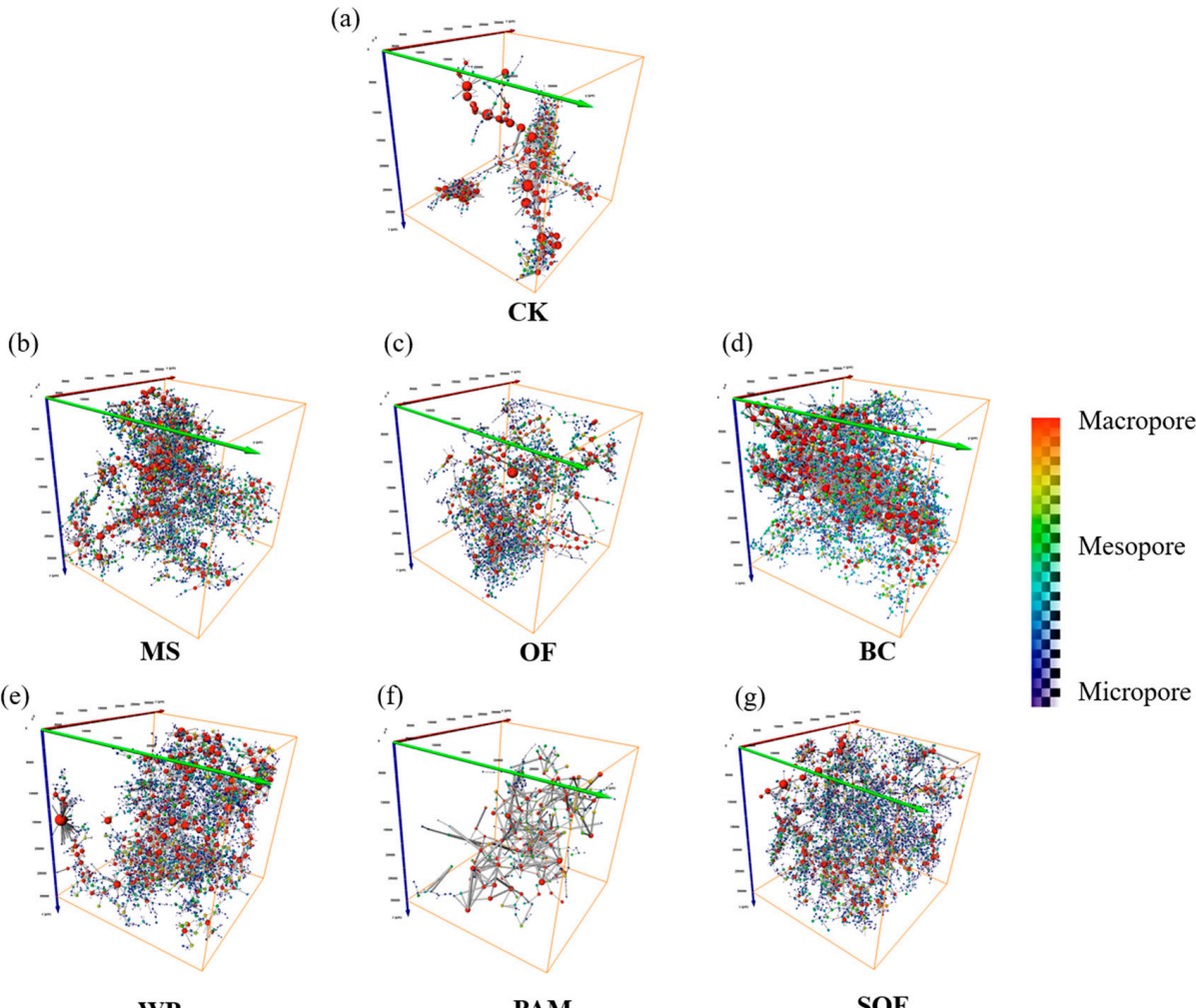

**Figure 5.** Representative 3D visualizations of soil pore networks under different treatments (physical size = 3 cm × 3 cm × 3 cm). Spheres represent node pores, and lines represent their connections. Warmer colors represent larger volumes, and cooler colors represent smaller volumes. (**a**) CK, with control, no organic material incorporated; (**b**) MS, maize straw; (**c**) OF, organic fertilizer; (**d**) BC, biochar; (**e**) WP, woody peat; (**f**) PAM, polyacrylamide; (**g**) SOF, maize straw and organic fertilizer.

**Table 5.** Quantification of the pore network of the soil cores with different organic treatments.

| PNM Properties | CK | MS | OF | BC | WP | PAM | SOF |
|---|---|---|---|---|---|---|---|
| Number of node pores | 2036.33 (485.91) | 3850.67 (1421.23) | 3412.67 (1047.07) | 4164.33 (1096.91) | 5522.67 (1406.10) | 4381.67 (2302.48) | 5989.67 (137.33) |
| Average coordination number | 3.11 (0.33) | 3.15 (0.33) | 3.26 (0.35) | 3.32 (0.18) | 3.03 (0.17) | 3.00 (0.13) | 3.60 (0.30) |
| Number of channels | 3059.33 (571.53) | 6530.67 (2630.63) | 5903.33 (2255.33) | 7043.33 (2156.12) | 8547.00 (2472.10) | 6794.00 (3779.67) | 10,729.33 (672.77) |
| Average throat area ($\mu m^2$) | 161,371.02 (65,068.80) | 163,365.75 (22,747.30) | 101,363.51 (5032.82) | 159,728.80 (40,334.15) | 112,121.09 (12,509.22) | 92,502.95 (18,928.89) | 101,140.47 (15,554.35) |
| Average channel length ($\mu m$) | 1116.12 (86.99) | 1131.05 (32.60) | 1081.49 (69.03) | 1174.26 (29.63) | 1031.15 (43.37) | 1567.34 (584.13) | 1013.98 (44.53) |
| Tortuosity | 3.24 (0.08) | 3.28 (0.04) | 3.36 (0.11) | 3.21 (0.10) | 3.19 (0.01) | 3.22 (0.05) | 3.21 (0.05) |

Values in parentheses represent the standard error of the mean.

A more detailed quantitative description of the channels and throats in the PNMs is presented in Figure 6. As the channel lengths increased, the number of channels for all treatments tended to increase and then decrease, with an overall positively skewed distribution, and the peak occurred near 1000 μm. Considering channel lengths below 500 μm, WP had more channels than the other organic treatments. At channel lengths between 500 and 1500 μm, SOF had the most channels, and CK had the fewest. BC had more long channels > 1500 μm but did not have a significant advantage over the other treatments. As shown in Figure 6b, the overall distribution pattern of the throat areas was similar among the different treatments, while the number of throats among treatments varied significantly. The number of throats in organic-treated soil cores was greater than that of CK in all ranges. SOF, BC, and OF had the largest number of throats for a throat area of $10^4$ $\mu m^2$, while the maximum number of throats in the other treatments occurred at $10^5$ $\mu m^2$.

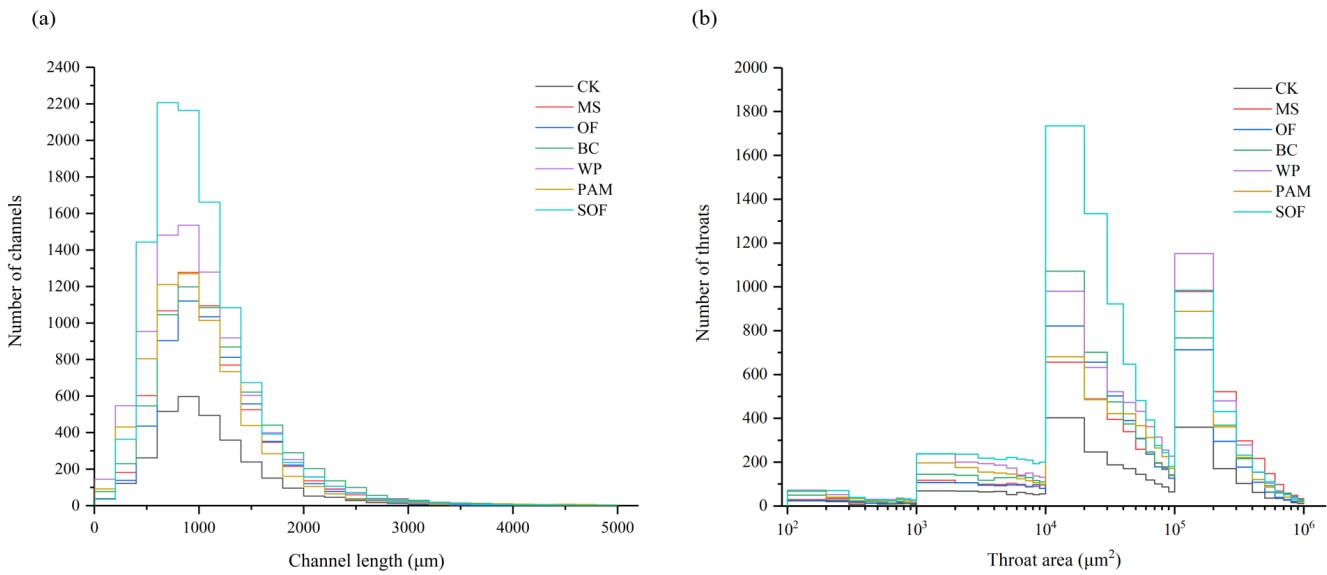

**Figure 6.** Distribution of channel length and pore throat area in the organic-treated soil cores. (**a**) channel length and (**b**) pore throat area.

### 3.5. Correlation of Extreme Precipitation Events and Pore Structure Characteristics of Treated Soil

The heatmap intuitively shows the correlation of soil pore characteristics for soil containing organic amendments and extreme precipitation events, and all the parameters are clustered into three groups (Figure 7). WP, BC, OF, and SOF were grouped with connectivity porosity, C/I ratio, and average coordination number, indicating that these treatments mainly affected the connectivity of soil cores, but the correlations did not reach a statistically significant level. The correlation between these indicators representing the connectivity of soil cores was significant. In the second group, the MS and average throat area were included, indicating that MS application mainly affected the throat of the PNM and that they were significantly positively correlated. PAM, extreme precipitation events, average channel length, and the properties related to pore shape were clustered into a group. The PAM treatment significantly increased the fraction of irregular pores as well as the average channel length. The extreme precipitation markedly increased the mean shape factor and significantly increased the isolated porosity.

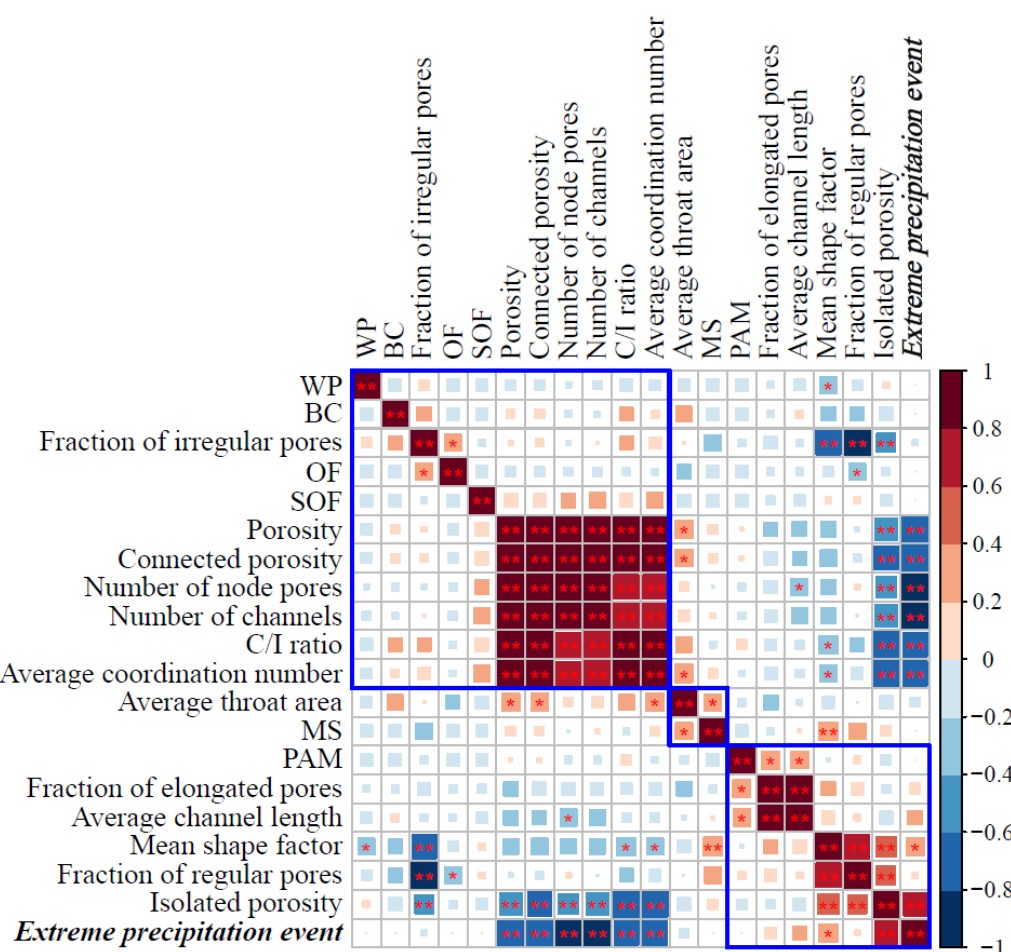

**Figure 7.** Correlation heatmap analysis for the soil pore characteristics of soils containing organic amendments and extreme precipitation events. The ordering method for the correlation matrix is hierarchical clustering based on Euclidean distance, according to which the parameters are divided into three groups (blue boxes in the figure). * and ** indicate correlation at $p = 0.05$ and 0.01, respectively. The correlation coefficients range from −1 to 1, with a value between −1 and 0 being negative and between 0 and 1 being positive.

In addition to the information within the clustered groups, the heatmap also indicated other correlations. There was a significant negative correlation between WP and the mean shape factor. The application of OF can significantly increase the fraction of irregular

pores and correspondingly decrease the fraction of regular pores. The isolated porosity was significantly positively correlated with both the mean shape factor and the fraction of regular pores, suggesting that the isolated pores within the soil cores may be closer to spherical. Extreme rainfall and porosity, connected porosity, number of nodes and channels, and C/I ratio were all strongly negatively correlated, which was similar to isolated porosity. This indicates that extreme precipitation significantly increased isolated porosity and destroyed the connectivity of the soil.

## 4. Discussion

### 4.1. Comparison of the Effect of Organic Amendments on Soil Pore Structure under Different Rainfall Conditions

This study showed that the test area suffered from extreme rainfall erosion during the 2021 maize season and that the soil water content of the cultivated layer was almost fully saturated following the rainfall events. On this basis, the extreme precipitation events, in addition to not affecting the proportion of pore space of each shape, significantly reduced the soil porosity and destroyed the connectivity of the soil. This agreed with the results of Panini et al. [31], and Todisco et al. [32]. However, it was also noted that this decrease in porosity was neither monotonous nor continuous, and may increase during the inter-rainfall period to counteract the effect of rainfall on porosity. More data is needed in subsequent studies may better assess the relationship between soil resilience and continuous rainfall in terms of porosity.

The introduction of organic materials mitigated the negative effects that extreme weather had, in varying degrees, on the pore parameters. Specifically, organic treatments increased soil porosity but not isolated porosity, and increased soil connectivity (Table 4). In this study, BC offered the most significant improvement. Under waterlogging stress, BC had a higher C/I ratio than the other treatments, implying better connectivity in the biochar-treated soil, which is consistent with previous studies [33–35]. In addition, such decisions on the incorporation of biochar should be made in conjunction with information about the effects of biochar on soil biochemical characteristics.

The analysis of the pore distribution characteristics showed that organic treatments increased the total porosity and connectivity, as measured in the soil core, but the proportion of pores of different sizes and shapes in the whole structure did not change. Previous studies have shown that organic amendments can change the pore size distribution within the soil, e.g., biochar increased the porosity of micropores and mesopores in soil, which was beneficial for nutrient utilization and water retention [36], PAM increased the number of mesopores and decreased the number of macropores [37], and the long-term application of dairy manure led to improvements in pore volume and size distribution [38]. Thus, we infer that extreme rainfall events may offset the effectiveness of organic amendments in terms of pore distribution. Previous studies provided theoretical support for the inference that water flow during the wetting process (rain or irrigation) may transport finer soil particles within coarser pores, thus averaging the fraction of each class of pore in the soil

In conclusion, the application of organic amendments, especially BC, has a positive effect on soil pore structure but is not sufficient to reverse the damage caused by heavy rainfall.

### 4.2. Effects of Organic Amendments on PNM under Waterlogging Stress

Pore network models and network analysis can be used to measure and estimate soil gas diffusivity [39], present the characteristics of the preferential flow and wetting front during the water-heat transport process [40], and predict many other physical properties of soils [41,42]. In our study, each of the organic modifiers could increase the complexity of the PNMs as evidenced by an increase in the number of node pores, channels, and throats. More node pores in PNMs are usually connected to more neighborhood pores, leading to a higher coordination number, which implies better connectivity within the soil. Among all organic treatments, BC and SOF were the most prominent, which may imply that these two organic amendments can improve the water retention capacity of the soil and have positive

effects on plant-available and hygroscopic water, as has been demonstrated by the results of related studies [43–45].

In addition to predicting the abovementioned soil properties, PNMs have also been studied to summarize some of the properties that facilitate soil solute transport and plant root growth. This includes determining (a) a low channel number, (b) a long channel length, (c) a low throat number, and (d) a large throat area [46,47]. Unlike the previous season which was optimal for PAM [27], the MS treatment created the best environment for fluid flow and crop root distribution during flooding as compared to those in the CK treatment. This result is similar to the findings of Chen et al. [48], He et al. [49], and Paul et al. [50], indicating that under extreme precipitation conditions, prolonged application of straw is more beneficial to crop growth than short-term quantitative spraying of PAM.

Taken together, the application of organic materials can complicate the pore network structure: BC–treated soil and SOF-treated soil have better connectivity, while MS can optimize the soil structure for crop growth. The modeling and analysis of the pore network can be used to predict the hydraulic properties and other related characteristics of organically amended soils, and its reliability should be verified by laboratory tests in future studies.

*4.3. Response of Pore Properties to Organic Amendments and Extreme Precipitation Events*

In this study, heatmap analysis was applied to investigate the correlation between extreme precipitation events and the pore structure characteristics of organically treated soils. The results were then grouped using hierarchical clustering based on Euclidean distance. WP, BC, OF, and SOF were clustered with connectivity porosity and C/I ratio, indicating that these four organic treatments mainly affected connectivity properties. The differences were that WP and OF showed negative correlations with the connectivity indicators, while BC and SOF showed positive correlations, but none of these correlations were significant. This result may imply that organic fertilizers alone do not promote soil connectivity as well as the combination of straw and organic fertilizers. The application of MS mainly affected the average throat area, and it was positively correlated with both the throat area and the mean shape factor. The larger the laryngeal tract area and mean shape factor are, the more favorable the transport of water and solutes [51], which is consistent with the results discussed above. PAM significantly increased the length of the channels and thus the fraction of elongated pores, indicating that the treatment provided a more beneficial environment for plant root growth [47]. However, this phenomenon was not manifested after exposure to heavy rainfall-driven erosion.

In addition to the usage of organic amendments, there was a strong correlation between extreme precipitation events and the soil pore parameters. Extreme rainfall was strongly negatively correlated with all indicators, which indicated good connectivity but significantly increased the isolated porosity within the soil. This analysis reveals that the damage to soil connectivity by waterlogging hazards is substantial and thus may have an impact on crop production, which should be further verified by future studies.

Overall, both organic material application and waterlogging damage were significantly associated with most soil pore structure parameters. However, the role of the organic materials was mainly in improving the structure, while extreme rainfall events were detrimental to soil connectivity.

## 5. Conclusions

This study indicated that extreme precipitation events can substantially reduce the effects that organic additives may have on the improvement of soil pore structure. Of all amendments, biochar was the most resistant to erosion and the most effective in improving porosity and connectivity. Our study further revealed that soil pore structure characteristics responded differently to organic amendments under different rainfall conditions. During conventional rainfall, the application of SOF mainly improved soil connectivity, PAM was conducive to the redistribution of water and nutrients in the soil, and MS mainly changed pore shape. However, during extreme rainfall events, the organic amendments affecting

the above parameters changed to BC, MS, and PAM, respectively. The combination of X-ray microtomography and advanced digital image analysis methods provided powerful tools for evaluating the effect of organic amendments on the structure of soil subjected to waterlogging stress at the microscale. In the future, the relationship between successive rainfall and the soil's ability to recover should be further investigated, and appropriate structural improvement measures should be selected according to the characteristics of the organic amendments and the physicochemical properties of the soil.

**Author Contributions:** Conceptualization, K.X., J.L. and J.Z.; methodology, K.X., J.L. and X.L.; software, K.X. and B.M.; validation, K.X. and Y.J.; investigation, K.X. and Y.J.; resources, K.X. and Y.J.; data curation, K.X.; writing—original draft preparation, K.X.; writing—review and editing, K.X. and J.L.; visualization, K.X.; supervision, J.L. and X.L.; funding acquisition, J.L., X.L. and J.Z. All authors have read and agreed to the published version of the manuscript.

**Funding:** This research was funded by the Strategic Priority Research Program of the Chinese Academy of Sciences (Grant No. XDA28010400), the National Key Research and Development Program of China (Funding number, 2022YFD1500502) and the National Natural Science Foundation of China (Funding number, 42177302).

**Data Availability Statement:** The data presented in this study are available on request from the corresponding author.

**Conflicts of Interest:** The authors declare no conflict of interest.

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
