# Peer review of "Effects of Organic Amendments on Soil Pore Structure under Waterlogging Stress"

_agronomy, doi:10.3390/agronomy13020289_

Round 1
Reviewer 1 Report
Dear authors! Thank you very much for a well written article. I don't see the point in making significant changes to the text. While reading the material there are comments of a recommendatory nature. For example, I consider the phrase "However, few amendments and porter characteristics were significantly correlated" (29) in the abstract is superfluous. The "above mentioned" is written as "above mentioned" (40). Therefore, I ask the authors to carefully re-read the text again. I wish authors success!
Author Response
Thank you for your supportive suggestions and we appreciate the opportunity to revise and resubmit our manuscript entitled "Effect of organic amendments on soil pore structure under waterlogging stress" (agronomy-215168). Your comments were valuable and constructive in improving our manuscript. We have carefully studied all comments and deleted "However, few amendments and porter characteristics were significantly correlated" in the manuscript and changed "abovementioned" to "above- mentioned".
Thank you again for your comments and suggestions.
Reviewer 2 Report
The MS is well presented but I think it needs improvement in the introductory section where there is only a small mention of biochar, which was found to be the best of the soil improvers used, I think it needs its own separate paragraph. I suggest the authors to edit some of the graphs, which are not very understandable. Discussions well done.

Author Response
Thank you for your supportive advice and we appreciate the opportunity to revise and resubmit our manuscript entitled " Effects of Organic Amendments on Soil Pore Structure under Waterlogging Stress" (agronomy-2155168). Your comments are valuable and very constructive for improving our manuscript. We have studied all comments carefully, and corresponding discussions were added to the manuscript. A point-by-point response to Reviewer 2’s comments was provided in the uploaded Word document. For clarity, we present the reviewers’ comments in normal text and our response to comments in normal text (red).
Thank you again for your comments and suggestions.

Reviewer 3 Report
The manuscript entitled ''Effects of Organic Amendments on Soil Pore Structure under Waterlogging Stress'' is a good paper with an interesting subject about soil amendments that potentially can be published. Research like this in soil structure on a field scale is worth consideration! Overall it is well written with a good interpretation of findings. But there are some small points that need to be improved. Please find some comments in the pdf.
Good Luck!

Author Response
Thank you for your supportive advice and we appreciate the opportunity to revise and resubmit our manuscript entitled " Effects of Organic Amendments on Soil Pore Structure under Waterlogging Stress" (agronomy-2155168). Your comments are valuable and very constructive for improving our manuscript. We have studied all comments carefully, and corresponding discussions were added to the manuscript. A point-by-point response to Reviewer 3’s comments was provided in the uploaded Word document. For clarity, we present the reviewers’ comments in normal text and our response to comments in normal text (red).
Thank you again for your comments and suggestions.
